# On Plasma Activated Acetyl Donors: Comparing the Antibacterial Efficacy of Tetraacetylethylenediamine and Pentaacetate Glucose

Endre J. Szili [1], Bethany L. Patenall [2], Adrian Fellows [3], Dharmit Mistry [3], A. Toby A. Jenkins [2], Robert D. Short [1,2,*] and Bhagirath Ghimire [4,*]

1    Future Industries Institute, University of South Australia, Adelaide, SA 5095, Australia
2    Department of Chemistry, University of Bath, Bath BA2 7AY, UK
3    AGA Nanotech Ltd., 2 Regal Way, Watford, Hertfordshire WD24 4YJ, UK
4    Department of Chemistry and Materials Science Institute, Lancaster University, Lancaster LA1 4YB, UK
*    Correspondence: r.d.short1@lancaster.ac.uk (R.D.S.); ghimirebhagi@hotmail.com (B.G.)

**Abstract:** The study compares how acetyl donor molecules tetraacetylethylenediamine (TAED) and pentaacetate glucose (PAG) improve the antibacterial efficacy of solutions activated with a low-temperature atmospheric-pressure argon plasma jet. Plasma activation of solubilised TAED and PAG produce solutions with different chemical compositions and oxidative potentials. Both acetyl donor molecules enhance the hydrogen peroxide ($H_2O_2$) concentration in solution with TAED being more effective compared to PAG. However, PAG is more effective at forming peracetic acid (PAA) from reaction of its acetyl donor groups with plasma generated $H_2O_2$. The enhanced oxidative potential of plasma activated TAED and PAG solutions were shown to significantly improve bactericidal activity against common wound pathogens Gram-negative *Pseudomonas aeruginosa* and Gram-positive *Staphylococcus aureus* compared to plasma activated water produced without acetyl donors. Furthermore, the oxidative capacity of plasma activated PAG was least affected by the bacterial oxidative defence enzyme catalase, attributed to the high concentration of PAA produced in this formulation. Overall, the above data show that acetyl donors may help improve next generation of antimicrobial formulations produced by plasma, which might help combat increasing problems of antimicrobial resistance.

**Keywords:** low-temperature atmospheric-pressure argon plasma jet; tetraacetylethylenediamine; pentaacetate glucose; hydrogen peroxide; peracetic acid; antibacterial; antimicrobial resistance

## 1. Introduction

Antimicrobial resistance (AMR) is an increasing environmental and healthcare problem that threatens to become a global crisis if replacements to antibiotics are not found. The first reported case of antibiotic resistance was penicillin resistance in 1942, only one year after the drug was administered to the first patient. Since the 1900s there has been an escalation of reports to antibiotic resistance [1]. The problem is compounded by the ability of bacteria to develop resistance to antibiotics at a faster rate than new classes of antibiotics can be discovered. Since a productive period of discovery between 1940–1980, antibiotic discovery has stalled with the last antibiotic discovered decades ago in 1987. AMR is currently predicted to become the leading cause of death by 2050, with 10 million deaths annually equating to 1 death every 3 s, resulting in a potential US $100 trillion shock to the global economy [2]. Urgent strategies are required to replace antibiotics to avoid this humanitarian, environmental and economic disaster.

Plasma activated water (PAW) is a potential alternative to antibiotics for treatment of localised infections and for controlling AMR in healthcare, environmental and agricultural settings [3,4]. PAW is prepared by treating water with an electrically ionized gas discharge

(plasma) [5–7]. Plasma produces different oxidizing molecules including reactive oxygen and nitrogen species (RONS) in the water that kill microorganisms through oxidative stress [3,8,9]. After plasma treatment of the water, highly reactive molecules such as the hydroxyl radical ($^\bullet$OH) and nitric oxide (NO) produced by plasma, quickly react to produce hydrogen peroxide ($H_2O_2$), nitrite ($NO_2^-$) and nitrate ($NO_3^-$) molecules and decrease the pH (due to production of peroxynitrous acid) in PAW [6,10–12]. These three molecules are relatively stable in PAW that act in synergy at low pH to kill microorganisms [5].

One challenge of developing PAW as a versatile antibacterial agent is that its bactericidal efficacy can vary depending on the type of microorganisms [13]. For example, bacteria with highly efficient antioxidant defence mechanisms may release catalase in response to PAW treatment; an enzyme that neutralises $H_2O_2$ to protect against oxidative stress [14]. This is a problem because $H_2O_2$ is a major antibacterial component in PAW [15]. To overcome this problem, we recently developed a method using an acetyl donor tetraacetylethylenediamine (TAED) to amplify the oxidative capacity of PAW [16,17]. During plasma treatment of the water, plasma produced $H_2O_2$ activates the acetyl donor to produce a potent oxidant peracetic acid (PAA) as previously described [1]. PAA is an ideal antimicrobial agent in PAW as there have been no reported incidence of PAA resistance, it decomposes to biocompatible and environmentally harmless agents (acetic acid, oxygen and water), and the acetyl donor precursors can be manufactured using green chemistry. We discovered that this approach produced a highly potent antibacterial solution that was resistant to catalase and was effective at reducing the growth of bacteria (*Pseudomonas aeruginosa* and *Staphylococcus aureus*) and at inactivating a virus (SARS-CoV-2) [17].

The motivation behind the present study is to evaluate if the antimicrobial efficacy of plasma activated solutions can be further improved with another acetyl donor pentaacetate glucose (PAG). A possible of advantage of using PAG over TAED in PAW is that PAG contains an additional acetyl donor group that might improve the efficiency in the generation of PAA.

## 2. Experimental Methodology

### 2.1. Plasma Jet Setup

The experimental methods were designed to enable a direct comparison between plasma activation of TAED, PAG and deionised water (DIW) without acetyl donors. The effect of the acetyl donors was followed at all stages of the research from plasma activation of the solutions through to how this impacted upon their antibacterial efficacy. The comparison involved assessment of electrical and optical characteristics of the plasma discharge, chemical analysis of the plasma activated solutions and bacterial response using common wound pathogens.

The schematic of the plasma device and overall experimental setup utilized in the study are shown in Figure 1a,b, respectively. The plasma jet used for the study is equipped with a stainless steel electrode (inner diameter, ID = 0.6 mm; outer diameter, OD = 0.9 mm) inside a quartz tube (ID = 1.5 mm; OD = 3.0 mm). The stainless steel electrode served as the high voltage (HV) electrode. The ground electrodes were made of copper tape (length = 4 mm) and fixed to the outer wall of the quartz tube at positions of 56 mm and 110 mm below the tip of the HV electrode. The length of the quartz tube below the end of the second ground electrode was 46 mm. The plasma jet configured in this way was chosen because it was previously shown to be optimal for efficiently producing $H_2O_2$ at room temperature [17,18]. The plasma jet was operated by purging argon gas at a flow rate of 1 standard litres per minute (SLPM) through the quartz tube and applying 3.56 kV (rms) at 23.5 kHz to the HV electrode using a sinusoidal power supply (PVM-500, Information Unlimited, P.O. Box 716, Amberst, NH 03031, USA). Voltage and current waveforms were measured at the HV electrode using a Pintek HVP-30 pro high voltage probe (Pintek Electronics Co. Ltd., Taipei, Taiwan) and a Pearson current probe (Pearson Electronics Inc., Palo Alto, CA, USA, model 2877), respectively, with the waveforms recorded on a Siglent SDS1102CML oscilloscope (Siglent Technologies Co. Ltd., Solon, OH, USA). The optical

emission signals of the discharge were recorded with an Ocean Optics HR4000CG-UV-NIR spectrometer (Ocean Optics Inc., Dunedin, FL, USA) with an optical fibre of diameter 600 μm.

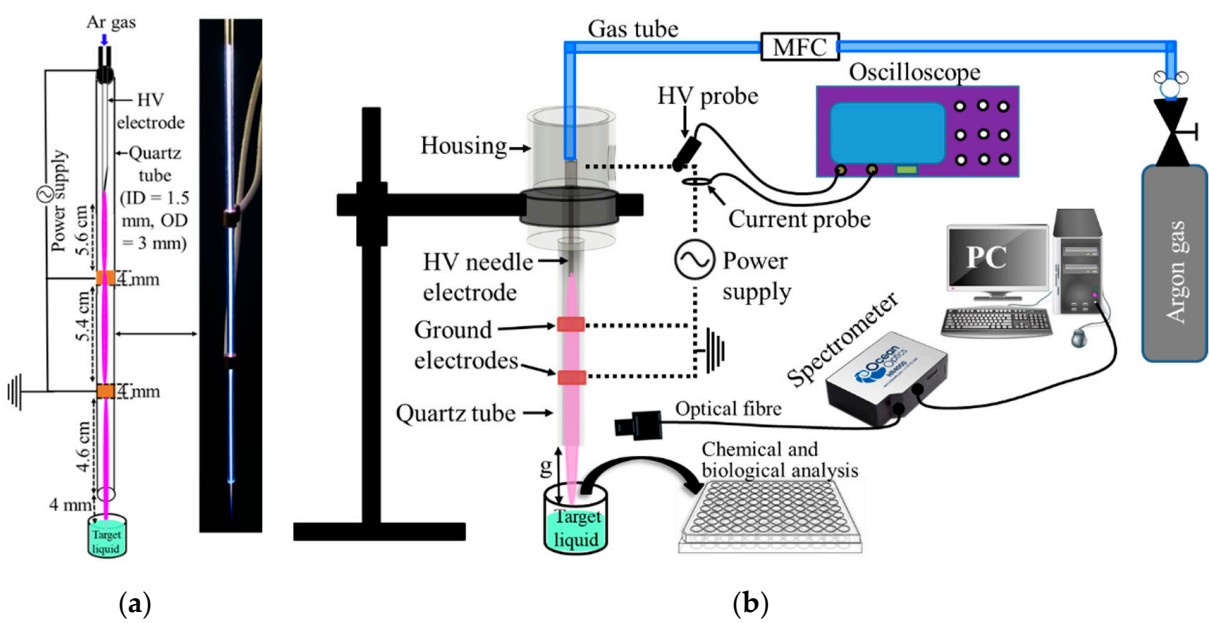

**Figure 1.** (**a**) Schematic and photograph of the plasma jet system used for the activation of acetyl donors; (**b**) schematic of the overall experimental setup.

## 2.2. Plasma Activated Solutions

The plasma activated solutions were prepared by activating TAED, PAG and deionised water (DIW) with the plasma jet, herein referred to as plasma activated (PA)-TAED, PA-PAG and PA-DIW. A 350 μL volume of 2.5 mM TAED (CAS number: 10543-57-4, Sigma Aldrich, St. Louis, MO, USA), 2.5 mM PAG (CAS number: 604-68-2, Sigma Aldrich, St. Louis, MO, USA) or DIW was dispensed in a well of a 96-well plate and treated with the plasma jet for 5 min at a 3 mm distance between the end of the quartz tube and water surface. During treatment, the plasma jet always remained in contact with the surface of the liquid.

## 2.3. Chemical Analysis of PA Solutions

The concentrations of $H_2O_2$ and PAA in the plasma activated solutions were measured to compare the activation efficiency of acetyl donors by plasma jets. $H_2O_2$ and PAA are the two major oxidants produced in the plasma activated solutions supplemented with acetyl donors and hence serve as a good indicator of their potential bactericidal activity as previously discussed [2].

$H_2O_2$ concentration in the plasma activated solutions was determined by measuring its oxidation of o-phenylenediamine (OPD, CAS number: 95-54-5, SigmaAldrich corporation, St. Louis, MO, USA) in the presence of horseradish peroxidase (HRP, CAS number: 9003-99-0, Sigma Aldrich Corporation, St. Louis, MO, USA). HRP catalyses the oxidation of OPD in the presence of $H_2O_2$ to form 2-3 diaminophenazine with an absorbance maximum of 450 nm. The absorbance maxima was measured using a plate reader (Synergy LX, Biotek, Santa Clara, CA, USA). Reagents were prepared by dissolving OPD and 20 μL of 2 mg/mL HRP in 10 mL of DIW. $H_2O_2$ concentration was determined from a linear line-of-best calibration curve constructed using known concentrations of $H_2O_2$ (CAS number: 7722-84-1, Sigma Aldrich, St. Louis, MO, USA). This involved adding 5 μL of plasma activated solution or $H_2O_2$ inside a 96-well plate containing 195 μL of OPD/HRP solution followed by 15 min of incubation. The absorbance readings obtained with the plasma activated solutions were converted to $H_2O_2$ concentration using the linear regression equation from the calibration curve shown in Figure 2.

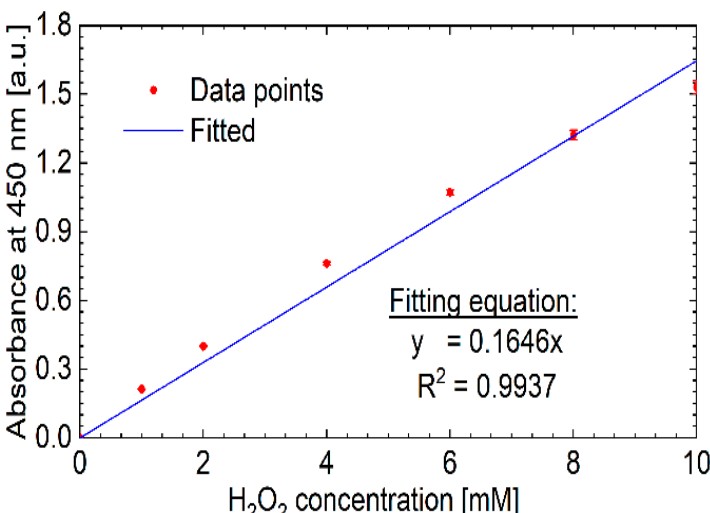

**Figure 2.** Calibration curve used for calculating the $H_2O_2$ concentration in the plasma activated solutions using the linear regression equation shown in the graph.

The concentration of PAA in the plasma activated solutions was determined by measuring its oxidation of potassium iodide (KI) using the following reagents: (a) KI (CAS number: 7681-11-0, Sigma Aldrich, , St. Louis, MO, USA), (b) catalase (CAS number: 9001-05-02, Sigma Aldrich, , St. Louis, MO, USA), (c) TAED, (d) PAG, and (e) PAA (Product code: 10302633, Fisher Scientific, , Loughborough, Leicestershire, UK). Catalase quenches $H_2O_2$ in the plasma activated solution leaving PAA as the major agent to oxidise KI as previously described [2]. A volume of 100 μL plasma activated solution or known concentrations of PAA (0.5 mM–8 mM used to prepare a calibration curve) were mixed with 100 μL of 75 mg/mL catalase in an Eppendorf tube and incubated for 15 min. Afterwards, 20 μL of the mixture was transferred to 180 μL of 100 mM KI and the solution was incubated for an additional 15 min before measuring the absorbance of oxidised KI at 350 nm. Absorbance readings obtained with the plasma activated solutions were converted to PAA concentration using the linear regression equation from the calibration curve shown in Figure 3.

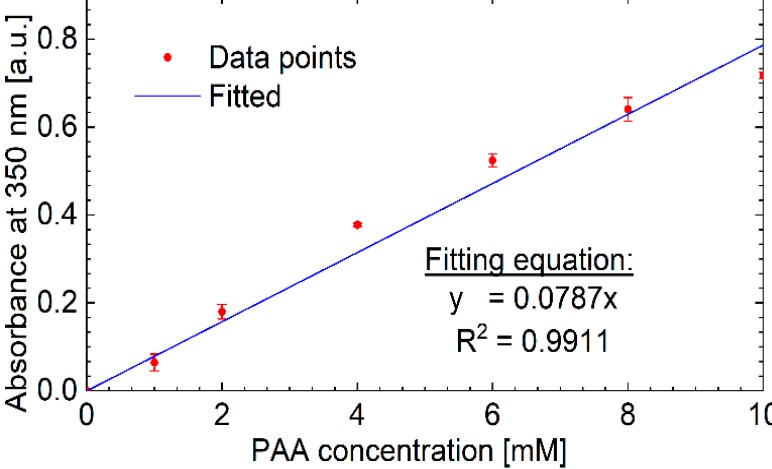

**Figure 3.** Calibration curve used for calculating the PAA concentration in the plasma activated solutions using the linear regression equation shown in the graph.

### 2.4. Oxidative Potential of Plasma Activated Solutions

The oxidative potential of the plasma activated solutions was determined by measuring their ability to oxidise KI in real-time. This was carried out by measuring the absorbance of oxidised KI at 350 nm using UV-vis spectroscopy. These measurements were performed immediately after plasma activation. A volume of 100 µL of the plasma activated solution or control solutions were transferred to a cuvette containing 500 µL of 100 mM KI and the corresponding UV-vis spectra were recorded after 1 min using a HR4000 spectrometer (Ocean Optics Inc.) equipped with a tungsten-halogen lamp. The temporal changes in the absorbance of the solutions were recorded for up to 15 min.

### 2.5. Antibacterial Efficacy of Plasma Activated Solutions

*Staphylococcus aureus* (H560) and *Pseudomonas aeruginosa* (PAO1) obtained from the Jenkins Collection at the University of Bath, were maintained in 15% (*v/v*) glycerol at −80 °C. As required, strains were streaked onto tryptic soy agar (TSA) and Luria-Bertani agar (LBA), respectively, to obtain single colonies. Single colonies were inoculated into 10 mL of tryptic soy broth (TSB) and Luria-Bertani (LB) broth, respectively, at 37 °C for 18 h. The cultures were centrifuged at 10,000 rpm and washed and resuspended in phosphate-buffered saline (PBS). Then, 10 mL of overnight culture was added into 10 mL of bacterial broth to make a subculture ($2 \times 10^6$ CFU/mL). The 100 µL aliquots of the bacteria subculture were added into wells of a 96-well plate followed by addition of 100 µL of the plasma activated solution (to achieve a 1:2 dilution). Further 1:2 dilution of the test solutions was carried out using the bacteria subculture solution to achieve a final dilution of 1:4. Bacteria were then grown statically for 18 h at 37 °C. Absorbance was recorded at 600 nm, which corresponded to solution turbidity which is indicative of bacterial growth. Bacterial growth was assessed relative to the negative control sterile broth as per the protocol outlined by the Clinical Laboratory Standards Institute (CLSI) [19].

### 3. Results

The results are set out to determine the differences and similarities in PA-TAED, PA-PAG and PA-DIW. This includes analysis of how TAED and PAG influences the properties of the plasma discharge and the resulting solution chemistry, followed by an assessment of how this correlates with the oxidative capacity of the plasma activated solution and finally how the differences in the characteristics of plasma activated solutions influence their antibacterial properties.

### 3.1. Current and Voltage Waveforms

Figure 4 shows typical current and voltage waveforms of the plasma jet operated at 3.56 kV (rms) at 23.5 kHz during plasma activation of the solutions. The waveforms during plasma activation of TAED, PAG or DIW solutions were similar. During the rising part of the applied voltage, charges were accumulated on the inner surface of the quartz tube resulting in the positive current peaks [12,20,21]. At least two current peaks were observed during the rising part of the applied voltage due to the presence of two ground electrodes in the plasma jet assembly [18]. These accumulated charges were reversed in polarity during the falling part of the applied voltage. The current (*I*) and voltage (*V*) values at every time point (*t*) were integrated over one time period (*T*) to estimate the maximum dissipated power [12,22]:

$$P = \frac{1}{T} \int_0^T I(t)V(t)dt \qquad (1)$$

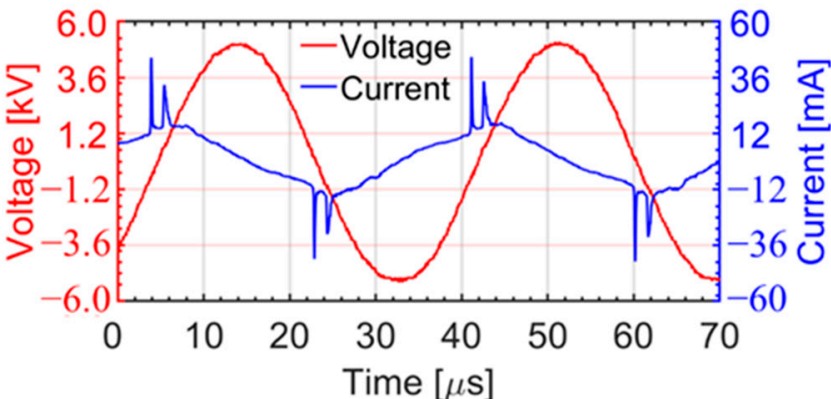

**Figure 4.** Current and voltage waveforms of the plasma jet during plasma activation of DIW.

The average dissipated power obtained using Equation (1) for the plasma jet in contact with DIW, TAED and PAG solutions was 6.71, 6.70 and 6.65 W, respectively. This suggests that the power consumed during production of the three solutions was nearly the same.

### 3.2. Optical Emission Characteristics

The optical emission spectra of the discharge recorded inside the quartz tube of the plasma jet is shown in Figure 5. The emission inside the quartz tube is mostly from the $^\bullet OH$ at 309 nm, atomic oxygen (O) at 777 nm and excited argon species between 600–900 nm [12,20,23]. The two ground electrodes of the plasma jet facilitated homogenous emissions along the length of the 164 mm long quartz tube. This feature of the plasma jet enabled higher production of $H_2O_2$ required for activation of the acetyl donor molecules within the solutions. Emissions outside of the quartz tube also comprise emissions from excited nitrogen species (see reference [17]). Plasma activation was carried out at a relatively close distance of 3 mm between the end of the quartz tube and surface of the solution, which favoured $H_2O_2$ production in the solution as opposed to nitrogen derivatives (which increase upon mixing with ambient air).

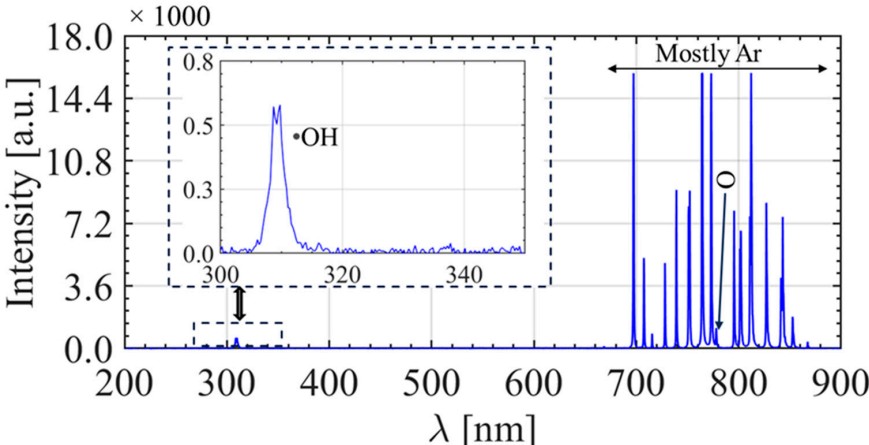

**Figure 5.** Optical emission spectra of the plasma jet recorded at 20 mm below the end of the second ground electrode (integration time = 70 ms).

### 3.3. Chemical Analysis of the Plasma Activated Solutions

Previously, it has been shown that the total concentration of RONS in PAW correlates with its absorbance profile measured by ultraviolet-visible (UV-vis) spectroscopy [24]. Therefore, as a first initial assessment, the UV-vis profile of the plasma activated solutions was measured. Figure 6 shows the UV-vis absorbance of PA-DIW, PA-TAED and PA-PAG across a wavelength range of 200–350 nm. Considering the area under the curves, it is seen that PA-TAED has a larger area of 192.80 square units, followed by PA-PAG of 102.32 square

units and lastly PA-DIW of 69.70 square units. Based on this comparison, we conclude that PA-TAED has the highest concentration of RONS, followed by PA-PAG and then PA-DIW.

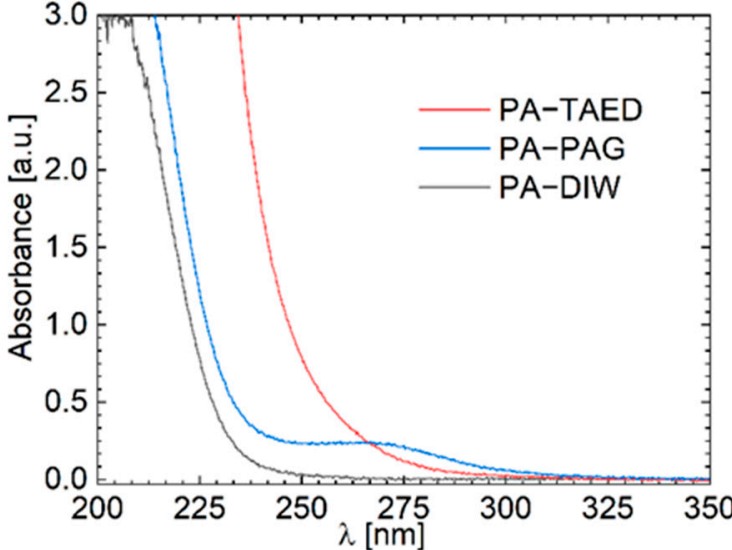

**Figure 6.** UV-vis spectra of the PA-DIW, PA-TAED and PA-DIW.

Based on our previous investigations, we determined that the main antibacterial efficacy of plasma activated acetyl donors is attributed to $H_2O_2$ and PAA [2]; however, we note that other molecules in the solution will also contribute to killing bacteria but are produced by plasma at much lower concentrations compared to $H_2O_2$ and PAA. Therefore, the concentrations of $H_2O_2$ and PAA produced in the plasma activated acetyl donor solutions was measured to obtain an indication of their potential antibacterial activity. Figure 7a,b shows the concentration of $H_2O_2$ and PAA, respectively, produced in PA-DIW, PA-TAED and PA-PAG. The highest $H_2O_2$ concentration was 5.0 mM produced in PA-TAED, followed by PA-PAG at 4.2 mM and PA-DIW at 2.5 mM. PA-PAG contained the highest PAA concentration at 6.3 mM compared to PA-TAED at 2.0 mM. As expected, PAA was not detected in PA-DIW due to the absence of acetyl donors. Figure 8 shows the pH of the plasma activated solutions. Addition of acetyl donors increased solution acidification with the lowest pH detected in PA-PAG.

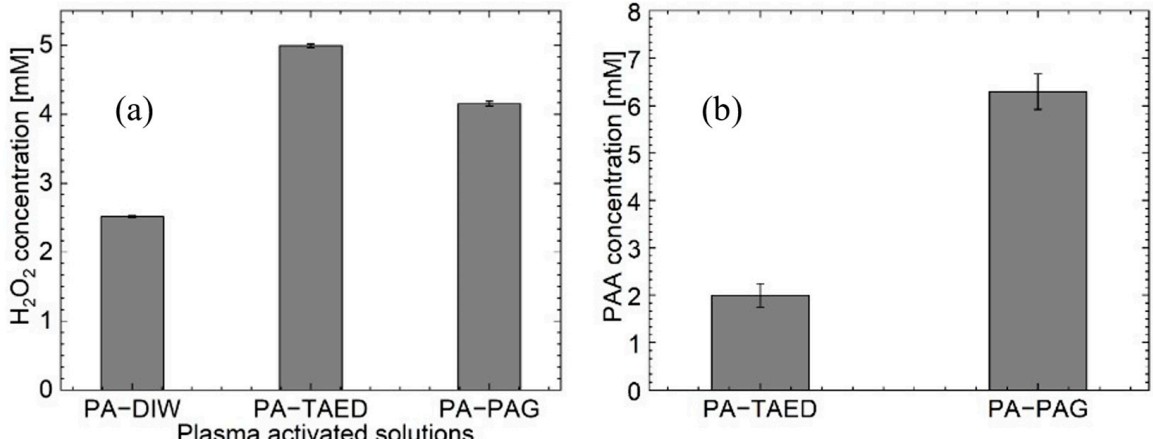

**Figure 7.** Concentrations of (**a**) $H_2O_2$ and (**b**) PAA produced in PA-DIW, PA-TAED and PA-PAG. Corresponding calibration curves used to calculate the concentrations are shown in Figures 2 and 3.

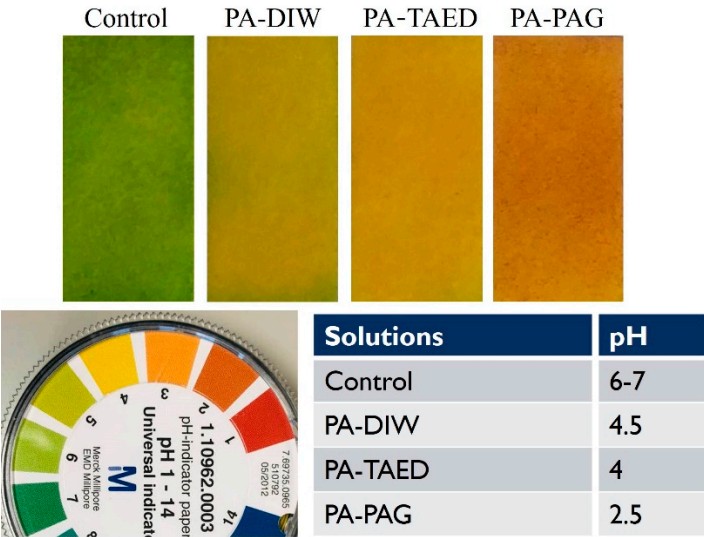

**Figure 8.** pH of PA-DIW, PA-TAED and PA-PAG in comparison to untreated DIW (Control).

### 3.4. Oxidative Capacity of the Plasma Activated Solutions

The oxidative capacity of the plasma activated solutions was measured by monitoring their ability to oxidise KI [25]. Both $H_2O_2$ and PAA will readily oxidise KI to produce yellow-coloured triiodide ions that can be measured using UV-vis spectroscopy at 350 nm (Figure 9a). The peak at 350 nm was continuously monitored by UV-vis spectroscopy up to 15 min during incubation of the plasma activated solutions with KI. The temporal changes in the UV-vis absorbance peak at 350 nm are shown in Figure 7b. It is seen in the graph that KI was oxidised fastest by PA-TAED, followed by PA-PAG and then PA-DIW. The oxidation of KI plateaued to the same level for PA-TAED and PA-PAG, and to a higher level compared to PA-DIW.

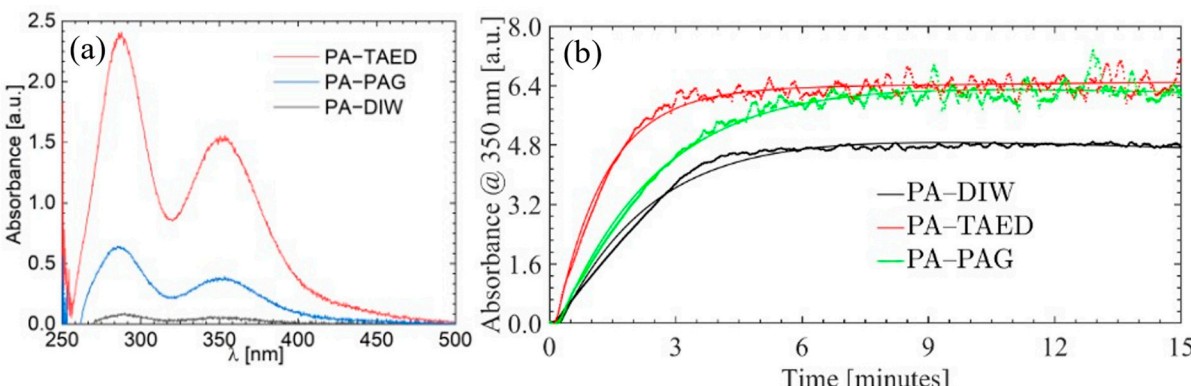

**Figure 9.** Oxidation of KI by PA-DIW, PA-TAED and PA-PAG as measured by UV-vis spectroscopy. Typical UV-vis profiles are shown in (**a**) indicating the peak for the triiodide ion at 350 nm used to (**b**) track the temporal changes in KI oxidation.

To understand how the bacterial antioxidant enzyme catalase affects the oxidative capacity of the plasma activated solutions, the oxidation of KI by PA-DIW, PA-TAED and PA-PAG was monitored with and without the presence of catalase. Figure 10 shows that catalase reduced the amount of KI oxidised by the plasma activated solutions due to the enzyme quenching $H_2O_2$. However, PA-PAG was affected the least by catalase (49% reduction in KI oxidation efficiency). PA-TAED was more sensitive to catalase (77% reduction in KI oxidation efficiency) whereas, PA-DIW was highly sensitive to catalase (92% reduction in KI oxidation efficiency).

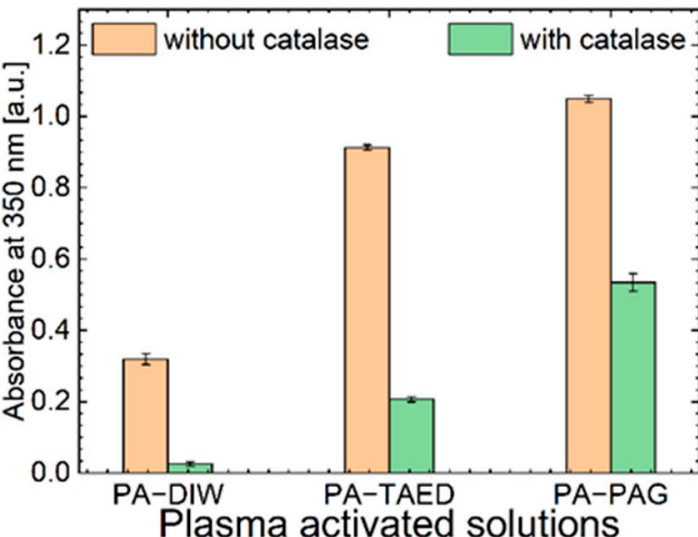

**Figure 10.** Oxidation of KI by PA-DIW, PA-TAED and PA-PAG with and without the presence of catalase.

To confirm if PAA helped improve the oxidative capacity of PA-TAED and PA-PAG in the presence of catalase, the oxidation of KI by $H_2O_2$ and PAA alone with and without the presence of catalase was plotted up to a relatively high $H_2O_2$ concentration of 10 mM and PAA concentration of 8 mM. It is seen that catalase completely abolishes the oxidative potential of $H_2O_2$ (Figure 11a). In comparison, PAA retains a relatively high oxidation potential in the presence of catalase (Figure 11b).

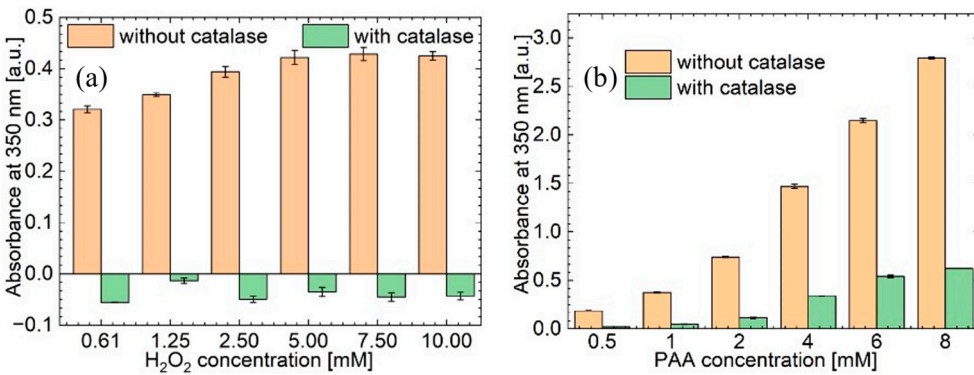

**Figure 11.** Oxidation of KI with different concentrations of (**a**) $H_2O_2$ and (**b**) PAA with and without the presence of catalase.

### 3.5. Antibacterial Efficacy of the Plasma Activated Solutions

The antibacterial efficacy of the plasma activated solutions was assessed using common wound pathogenic bacteria *P. aeruginosa* and *S. aureus* in the planktonic state. Bacterial growth was monitored by measuring the turbidity of bacterial broth at an OD of 600 nm. Figure 12 shows that PA-TAED and PA-PAG completely inhibited growth of bacteria (i.e., the $OD_{600nm}$ was the same for the bacterial suspensions after treatment with these plasma activated solutions compared to the sterile broth negative control). Conversely, PA-DIW was less effective at inhibiting bacterial growth.

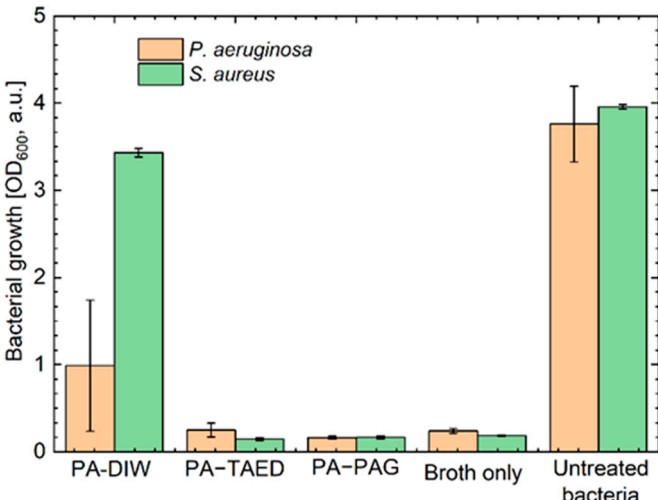

**Figure 12.** Growth of *P. aeruginosa* and *S. aureus* measured after treatment with PA-DIW, PA-TAED and PA-PAG. Sterile broth and untreated bacterial suspensions were used as negative and positive controls of bacterial growth, respectively.

## 4. Discussion

A few important differences between PA-DIW, PA-TAED and PA-PAG were discovered in this study. Both acetyl donors (TAED and PAG) enhanced the plasma production of RONS in the DIW. TAED facilitated the highest $H_2O_2$ concentration in the plasma activated solution in this study. However, addition of PAG resulted in the highest concentration of PAA and consequently lowest pH in the plasma activated solution. This result was attributed to the PAG molecule containing more acetyl donor groups that can be converted into PAA compared to TAED. It is unlikely that the acetyl donors themselves influenced the plasma jet production of RONS because the electrical and optical characteristics of the plasma discharge during plasma activation were similar for all solutions. Therefore, the differences in the plasma activated solution chemistry observed in this study are mainly attributed to the downstream chemical reactions in the DIW that were influenced by TAED and PAG. Briefly, the short-lived •OH produced within the core of the plasma jet and near the vicinity of the DIW's surface recombine to form $H_2O_2$ through the recombination reaction: •OH + •OH → $H_2O_2$ [17,26]. The generated $H_2O_2$ then reacts with acetyl donor molecules to form PAA [27]. The reaction of $H_2O_2$ with the acetyl donors generates additional $H_2O_2$ molecules as by-products which can increase the $H_2O_2$ concentration initially generated by the plasma. This was observed in the present study where PA-TAED and PA-PAG contained higher $H_2O_2$ concentrations compared to PA-DIW.

The acetyl donors also enhanced the oxidation potential of the plasma activated solution compared to PA-DIW. PA-TAED was shown to oxidise KI at a faster rate compared to PA-PAG. This result is due to the overall higher RONS concentration in PA-TAED. Although the oxidation rate of KI was faster with PA-TAED in the first few minutes of the reaction, when the reaction was allowed to reach completion, it was found that PA-TAED and PA-PAG oxidised KI to the same level, which was notably higher than what could be achieved with PA-DIW. Therefore, one major advantage of the acetyl donors is that both enhance the oxidative potential of the plasma activated solution.

The major advantage of producing PAA in the plasma activated solution is that it is resistant to catalase. It was shown that in the presence of catalase, PA-TAED and PA-PAG could retain a high level of oxidative activity, whereas the oxidative activity for PA-DIW was almost completely abolished. Further to this point, PA-PAG was shown to be more resistant to catalase compared to PA-TAED because it contains a higher concentration of PAA. This is an important result because further increasing the $H_2O_2$ concentration in PAW cannot help overcome its resistance to catalase, whereas PAA in PA-TAED and

PA-PAG significantly improves the oxidative capacity of the plasma activated solution in the presence of catalase.

The ability of plasma to easily produce a complex mixture of antibacterial agents in solution opens a myriad of new opportunities to developing new antimicrobial solutions for a range of applications including wound care, agriculture, food handling and dentistry. In developing plasma activated solutions for these applications, we should be cognisant of the likelihood that microbes may also develop resistance. Therefore, it is necessary to continue developing new plasma activated solutions to combat AMR. This study has shown that the type of acetyl donor used to produce the plasma activated solution can change its oxidative potential and increase its resistance to catalase, which is a major antioxidant defence enzyme produced by many bacteria. Both acetyl donors TAED and PAG investigated in this study significantly improved the antibacterial efficacy of the plasma activated solution. Although the use of both acetyl donors resulted in a similar antibacterial effect, it is highly likely that the bactericidal efficacy between plasma activated TAED and PAG might differ for other types of bacteria. Therefore, it is important to continue tailoring and optimising plasma activated formulations for each application to mitigate the risks of AMR.

An additional question which we did not specifically address is in this study is whether the plasma activated solutions are safe for clinical use. Plasma treatment can produce a stimulatory effect and inhibitory effect on cells depending on the plasma treatment time (which correlates with dose of reactive oxygen species) [28]. When a wound is infected, it is imperative to stop the infection from spreading particularly to prevent sepsis or osteomyelitis. In the plasma decontamination of a wound, it is reasonable to assume that some surrounding cells will be killed or removed (as we have previously noted in our work [18]); and this is a standard effect in the debridement and cleansing of wounds [29]. The minimum inhibitory concentration (MIC) of $H_2O_2$ differs for different bacterial strains; e.g., for common wound pathogens *P. aeruginosa* has a reported $H_2O_2$ MIC of (0.7–1.4 mM) whereas it much higher for *S. aureus* at 3–6 mM [30]. The concentrations we investigated in this study fall within these ranges. $H_2O_2$ use within wound care can be contentious, with high concentrations like the 3% (980 mM) solutions used in the clinic reported to delay healing, yet lower concentrations of up to 10 mM are reported to promote wound healing in vivo [31]. Importantly, the $H_2O_2$ concentration range used in our study falls into the latter. Based on the results in our study and the reported MICs of both bacteria we can conclude that our treatment is antibacterial and can potentially promote wound healing. Combining chemical species produced by plasma and peracetic acid provides a superior antibacterial solution compared to any of the agents used alone. The multipronged chemical application of the combined formulation improves bactericidal activity, which should help mitigate antimicrobial resistance.

## 5. Conclusions

Use of two acetyl donors TAED and PAG were shown to enhance the antibacterial effectiveness of the plasma activated solution. It is likely that this result is attributed to both acetyl donors raising the oxidation potential of the plasma activated solution and producing PAA that is resistant to an antioxidant enzyme catalase, which is released by bacteria in response to oxidative stress. A major advantage of using acetyl donors in the production of plasma activated solutions is that the molecules provide an environmentally friendly method of improving the antibacterial efficacy of the solution that may help in the future development of new antimicrobial formulations that are able to mitigate AMR in healthcare and in our environment.

**Author Contributions:** Conceptualization, all authors; methodology, all authors; validation, E.J.S., B.L.P. and B.G.; formal analysis, E.J.S., B.L.P. and B.G.; investigation, all authors; resources, A.F., D.M., A.T.A.J. and R.D.S.; data curation, E.J.S., B.L.P. and B.G.; writing—original draft preparation, E.J.S. and B.G.; writing—review and editing, all authors. All authors have read and agreed to the published version of the manuscript.

**Funding:** This research was funded by: EPSRC (Grant Nos. EP/R003556/1 and EP/V00607X/1), Australian Research Council Future (Fellowship No. FT190100263), the National Health Medical Research Council Ideas (Grant No. 2002510), and the Future Industries Accelerator Mobility Scheme MOB024.

**Institutional Review Board Statement:** Not applicable.

**Informed Consent Statement:** Not applicable.

**Conflicts of Interest:** The authors declare no conflict of interest.

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
