# Peer review of "On Plasma Activated Acetyl Donors: Comparing the Antibacterial Efficacy of Tetraacetylethylenediamine and Pentaacetate Glucose"

_plasma, doi:10.3390/plasma5040031_

Round 1

Reviewer 1 Report

I have read the manuscript entitled “On plasma activated acetyl donors: comparing the antibacterial efficacy of tetraacetylethylenediamine and pentaacetate glucose”. The study described here is concerned with the generation of hydrogen peroxide in solution by atmospheric pressure plasma jet. The authors describe and claim the antibacterial properties of the plasma treated water. A key point for this work is the chemistry involving the activated acetyl donors that enhance the production of hydrogen peroxide. I find the manuscript interesting and well written, however, I do have the following questions/points that the authors should consider.

A description of the chemical mechanism of the hydrogen peroxide should be described.

The authors claim that this method is good for antibiotics and quote that it may be useful for treating wound healing, (p. 11) probably due to the ability to localise the plasma. However, there is no discussion that hydrogen peroxide also is toxic for any other cells, so its actually not very targeted for bacteria. Please comment. The label antibacterial is misleading!

The method used to measure “bacterial concentration” is very non specific, there are points in the manuscript (P.11) where the authors state that bacterial are being killed. This may be so, but there is no differentiation between live and dead cells. Optical density measurements are completely misleading for this work. They measure the turbidity only, and bacterial ‘concentration’ only indirectly. But the problem is that they are using an inappropriate measure and associating this with inactivation of bacteria. This is at best is misleading at worst is scientifically incorrect.

Reviewer 2 Report

In this work, the authors compare the activation efficiency of a low-temperature atmospheric-pressure argon plasma jet in the case of two donor molecules tetraacetylethylenediamine (TAED) and pentaacetate glucose (PAG) to improve bactericidal activity against common wound pathogens Gram-negative Pseudomonas aeruginosa and Gram-positive Staphylococcus aureus.

The manuscript presents a very interesting topic and is well presented and structured.

I have a few issues that I propose the author address before acceptance of the manuscript.

-          Figure 1 from the appendix can be introduced into the main text where the figure is cited.

-          Figure 10 from section Discussion should be moved to section Results, or the two sections can be grouped into one section.

-          How about intra-day repeatability and inter-day reproducibility of the assay?

Round 2

Reviewer 1 Report

The changes made are appropriate, I recommend publication.